# Dimorphism of HLA-E and Its Disease Association

**DOI:** 10.3390/ijms20215496

**Published:** 2019-11-04

**Authors:** Leonid Kanevskiy, Sofya Erokhina, Polina Kobyzeva, Maria Streltsova, Alexander Sapozhnikov, Elena Kovalenko

**Affiliations:** Shemyakin-Ovchinnikov Institute of Bioorganic Chemistry, Russian Academy of Sciences, 16/10, Miklukho-Maklaya St., Moscow 117997, Russia; sonya.erokhina@gmail.com (S.E.); polinafakobyzev@gmail.com (P.K.); mstreltsova@mail.ru (M.S.); amsap@mail.ru (A.S.); lenkovalen@mail.ru (E.K.)

**Keywords:** HLA-E, peptide repertoire, antigen presentation, NK cells, NKG2 receptors

## Abstract

*HLA-E* is a nonclassical member of the major histocompatibility complex class I gene locus. HLA-E protein shares a high level of homology with MHC Ia classical proteins: it has similar tertiary structure, associates with β2-microglobulin, and is able to present peptides to cytotoxic lymphocytes. The main function of HLA-E under normal conditions is to present peptides derived from the leader sequences of classical HLA class I proteins, thus serving for monitoring of expression of these molecules performed by cytotoxic lymphocytes. However, opposite to multiallelic classical *MHC I* genes, *HLA-E* in fact has only two alleles—*HLA-E*01:01* and *HLA-E*01:03*—which differ by one nonsynonymous amino acid substitution at position 107, resulting in an arginine in *HLA-E*01:01* (*HLA-E^R^*) and glycine in *HLA-E*01:03* (*HLA-E^G^*). In contrast to *HLA-E^R^,*
*HLA-E^G^* has higher affinity to peptide, higher surface expression, and higher thermal stability of the corresponding protein, and it is more ancient than *HLA-E^R^*, though both alleles are presented in human populations in nearly equal frequencies. In the current review, we aimed to uncover the reason of the expansion of the younger allele, *HLA-E^R^*, by analysis of associations of both *HLA-E* alleles with a number of diseases, including viral and bacterial infections, cancer, and autoimmune disorders.

## 1. Introduction

*HLA-E* is encoded by the major histocompatibility complex (MHC) locus and belongs to non-classical MHC clаss I (MHC Ib) genes. HLA-E protein shares high structural and functional homology with other MHC class I molecules. In particular, it consists of three MHC class I-like extracellular domains, and two of them, α1 and α2, form the peptide-binding groove; α3 is a membrane proximal domain. HLA-E is expressed at the cell surface only after its association with β2-microglobulin and antigenic peptide. It differs from classical (MHC Ia) proteins by limited polymorphism and a specific set of presented peptides [1,2].

Normally, HLA-E almost exclusively binds to the limited set of 9-mer peptides, derived from leader peptides of classical HLA-A, B, C, G proteins [2,3] with the consensus sequence of VM(A/P)PRT(L/V) (V/L/I/F)L. This excludes several HLA-B allotypes which contain a Thr or Ala residue instead of Met, a few HLA-C allotypes, and the leader peptides from HLA-F and HLA-E itself that do not match this motif [2,4,5]. Two main anchor residues in the nonamer are Met in the position 2 and Leu in the position 9 [2] (Table 1). The pathway of generating of such peptides is distinct from that of antigenic peptides intended for binding MHC Ia proteins. During translation, the leader peptides are cut off by Signal Peptidase and then cleaved again by Signal Peptide Peptidase, following by the release of ~14 residue-long signal peptide fragments into the cytosol. Therein they additionally are trimmed by proteasome, and then transported into endoplasmic reticulum (ER )lumen by transporter associated with antigen processing (TAP) protein [6]. At last, these peptides enter into the HLA-E peptide-binding groove. This interaction, as well as association of HLA-E with β2-microglobulin promotes the correct folding of the HLA-E complex and allows it to be expressed at the cell surface [6]. However, in some cases the repertoire of peptides presented by HLA-E can be changed. In particular, under cell stress HLA-E is able to present peptide derived from heat shock protein 60 (Hsp60) [7]. In addition, a number of pathogens, including bacteria (*Salmonella, Listeria monocytogenes, Mycobacterium tuberculosis*) and viruses (CMV, HIV, HBV, EBV, HCV, VACV), produce peptides that are able to associate with HLA-E or its mouse counterpart Qa-1 [8,9]. The peptide sequence influences its binding affinity to HLA-E, which in turn correlates with the surface expression level of HLA-E-peptide complex [10].

*HLA-E* is expressed on most nucleated cells of the human body. However, surface expression of HLA-E protein was found in a restricted set of tissues. In particular, it has been found on leukocytes, endothelium, and cells of trophoblast. The highest level of HLA-E expression was observed on the immune cells [4,13,14,15,16]. Expression of HLA-E is often detected on cells of different tissues in pathologic conditions, such as cancer, bone marrow transplantation, and autoimmune diseases [10,17,18,19,20].

Surface HLA-E–β2-microglobulin complex with antigenic peptide is detected by cytotoxic lymphocytes such as NK cells and a subset of CD8^+^ T cells via a group of receptors encoded by Natural Killer group 2 (NKG2) complex, such as NKG2A or NKG2C, and, in some cases, by αβTCR [21,22]. In this review, NKG2x/CD94 and αβTCR receptors will be discussed. Some of the aforementioned receptors, by being engaged with HLA-E, induce inhibitory signal that protects HLA-E- expressing cells from lysis, whereas others activate lymphocytes, leading to the elimination of target cells.

## 2. HLA-E and NKG2 Receptor Family

NKG2 receptor family is composed of seven members, namely NKG2A, B, C, D, E, F, and H. Cells which express HLA-E-peptide complexes can be recognized by cytotoxic lymphocytes via heterodimeric receptor complexes consisting of CD94 covalently bound with either NKG2A, NKG2B, or NKG2C. NKG2A and NKG2B proteins are isoforms derived from the same gene, *NKG2A,* by alternative splicing. NKG2B differs from NKG2A by the absence of the membrane-proximal fragment in the extracellular part of the molecule [23]. Both proteins have immunoreceptor tyrosine-based inhibition motif (ITIM) elements in their cytoplasmic tails and thus can provide the inhibitory downstream signals. Engagement of CD94/NKG2A or CD94/NKG2B with HLA-E-peptide induces inhibitory signal, thus preventing the lysis of the target cell by cytotoxic lymphocyte. By this mechanism, lymphocytes remain tolerant to autologous cells [3,24]. Concerning that, HLA-E normally presents the peptides derived from HLA-A, -B, -C (HLA-G is expressed only on a very restricted set of tissues [16]), and this mechanism of NKG2A/B-induced tolerance serves for monitoring of MHC class I molecules expression in the cells of the organism [24]. In the absence of MHC Ia expression, when there are no peptides to present, HLA-E becomes downregulated [5]. In this case, NK or T cells do not receive the inhibitory signal and can lyse the target cell using a number of activating receptors, which recognize a set of ligands expressed in conditions of infection, cell stress, or malignization of the cell.

NKG2C, opposite to NKG2A/B, is an activating receptor: Its engagement with HLA-E complexes leads to the cytolysis of the target cell [25]. It has two homologous proteins: NKG2E and NKG2H. These three receptors lack inhibitory ITIM elements in their cytoplasmic domains, but associate with CD94 and DAP12 proteins, the latter of which bearing activatory ITAM motif. NKG2E and NKG2H are the isoforms resulted from alternative splicing of one gene *NKG2E*. NKG2E has unknown functions, and only its intracellular localization has been determined [26]. NKG2H is expressed on the cell surface and can trigger activation, but not by binding with HLA-E. The ligand for NKG2H is still unknown. However, interestingly, the activation of NKG2H-expressing T cells has been shown to markedly reduce the activation and proliferation of other T cells in the culture [27]. It can be assumed that such cells may play a regulatory role.

Other receptors encoded by NKG2 locus include NKG2D and NKG2F. NKG2D is well described receptor for MIC and ULBP proteins [28]. NKG2F contains inhibitory ITIM motif, but can also associate with DAP12. This receptor is only found intracellularly, probably because of its inability to bind with CD94. Its functions are unknown; authors suggested that it can be a rudimentary gene product [29].

Two NKG2 receptors—NKG2A, an inhibitory receptor, and NKG2C, an activatory receptor—expressed on both NK and T cells will be discussed here. NK cell usually expresses only one type of receptors, either inhibitory or activatory, i.e., NKG2A or NKG2C. However, there is a small subset of NKG2A^+^NKG2C^+^ NK cells with unknown functions (Figure 1).

NKG2C is well-studied activating receptor member of NKG2 family, however, its role to date is still not yet clear. The research of the last two decades clearly showed that expression of NKG2C on NK cells is tightly associated with human cytomegalovirus (HCMV) infection [25,30,31]. This virus infects 50–100% of human population worldwide, however, in most cases it persists in the organism latently without clinical manifestations. Its spread is controlled by T and NK cells, which are unable to eliminate completely the infection from the human body. HCMV was shown to downregulate the MHC Ia expression, that allowed avoiding the recognition of the infected cells by T cells. Such cells, in theory, should be lysed by NK cells, which can recognize and kill MHC Ia-negative cells. However, HCMV encodes and produces the peptide derived from viral protein UL40, which is analogous to that of derived from the leader sequence of MHC class Ia molecule, which normally binds to HLA-E. This mechanism allows to retain the HLA-E expression on infected cells and thereby evade the NK cell-mediated lysis via NKG2A. However, there is a subpopulation of NK cells that have been shown to control the spread of HCMV. These cells bear the activating receptor NKG2C, which stimulates cytotoxicity and cytokine production by NK cells when engaged by complex of HLA-E with peptide derived from protein UL40 [32]. They possess the features of immunological memory, in particular, the clonal expansion and the enhanced response to the re-encountering antigen, and are currently called adaptive NK cells [31,33,34,35]. Summarizing the available data, it can be concluded that HCMV infection in the body seems to precede the appearance of adaptive NK cells [36].

Complexes of HLA-E bound with the peptide derived from HLA-G, but not other HLA molecules (this peptide is different by Phe residue in position 8, see Table 1), have been shown to significantly increase the NKG2C^+^ NK cell degranulation [11,37]. Normally, HLA-G expression is not detected in most of tissues [16], however, it can be induced by viral infections, including HCMV [11]. HLA-G is expressed on different tumors [38], and this can partially explain the antitumor activity of adaptive NK cells. Some strains of HCMV produce UL40 protein derived peptide, which has the same sequence as such of HLA-G, and this peptide induces potent response of NKG2C^+^ NK cells in terms of cytokine production and cytotoxicity [32]. It is important to note that distribution of HLA-E in leukocytes and endothelial cells corresponds to the sites of the primary infection and latent residence of HCMV [36]. Altogether, these reports show that HLA-G-derived leader peptide fragment or, in some cases, UL40-derived peptide in complex with HLA-E seems to serve as targets for NKG2C^+^ NK cells, leading to their activation and cytotoxicity. So, the presentation of HLA-G-derived peptide in complex with HLA-E seems to have a different role in comparison with peptides generated from HLA class Ia (A, B, and C) proteins: It causes activation of NKG2C^+^ NK cells, rather than inhibition of NKG2A^+^ NK cells [36].

At last, it should be noted that the expression of inhibitory NKG2A/B receptors in individual NK cell clones at the mRNA level was 3–5-fold higher than such of activatory NKG2C receptor [39]. Moreover, affinity of NKG2C protein to HLA-E complex is 6-fold lower than that of NKG2A [40]. Obviously, this reflects the need to support the self-tolerant state of the immune system.

## 3. HLA-E-Peptide Complexes and TCR

CD8^+^ T cells express on their surfaces inhibitory receptor NKG2A, and, in higher extent, activatory NKG2C receptor [41]. Nevertheless, they are able to recognize complexes of HLA-E with peptides, also via αβTCR as an activatory receptor. Complexes of HLA-E with peptides derived from viruses or bacteria, including human cytomegalovirus (HCMV), *Mycobacterium tuberculosis,* and *Salmonella enterica* can induce cytotoxic CD8^+^ T cell response [12].

Non-canonical HLA-E restricted CD8^+^ T cells can recognize UL40-derived peptide of HCMV loaded on HLA-E. About 30% of seropositive hosts were shown to manifest HCMV-specific T-cell response which controls the expansion of virus during the whole life. The amount of these non-conventional HCMV-specific T cells can reach to 38% of total blood CD8^+^ T lymphocytes. These cells are effector memory CD8^+^ T cells [42]. It can be suggested that in other 70% of HCMV^+^ hosts HCMV activity is under the control of NKG2C^+^ NK cells.

Besides HCMV, non-conventional T cells can recognize a number of mycobacterial peptide antigens, presented by HLA-E [43,44]. This subset of HLA-E-restricted T cells, recognizing mycobacterial peptides, was found to be significantly higher in patients with tuberculosis in comparison with healthy donors, resembling the situation with HCMV [45]. Analogous data has been obtained for *Salmonella enterica* [46].

HLA-E-restricted T cells lyse infected cells and by this limit the spread of virus or bacteria. Together, these data suggest that NK and T cell receptors play complex roles: They monitor the surface expression of HLA I class proteins by NKG2A receptor, which is expressed on both NK and T cells, and eliminate intracellular pathogens using NKG2C on NK cells or HLA-E-restricted TCR on T cells.

## 4. Evolution and Polymorphism of HLA-E

*HLA-E* seems to be the most conserved among the other human MHC class I genes: Its homologues have been found even in New World monkeys [47]. To date, 27 alleles of *HLA-E* gene have been discovered, however, most of them are either very rare or encode only non-functional proteins [12]. In fact, two main alleles, *E*01:01* and *E*01:03*, together account almost for 100% in human population. This fact indicates that *HLA-E* was being under the pressure of stabilizing selection for a long period of time, in strict contrary to HLA class Ia genes, which have evolved a wide range of alleles. These two *HLA-E* alleles are distributed worldwide in roughly equal proportions [1,3] (see also http://www.allelefrequencies.net/hla6006a.asp?hla_locus_type = Non-Classical, locus E), although in a certain country this proportion may be shifted towards one or another allele. Thus, we would simplistically consider that there are only these two alleles, and discuss HLA-E dimorphism.

The polypeptide products of these alleles differ by a single amino acid residue substitution in position 107; allele **01:01* (*HLA-E^R^*) has the Arg107, allele **01:03* (*HLA-E^G^*) – Gly107. This position is located outside of the peptide-binding groove of the HLA-E molecule; however, it affects some characteristics of HLA-E, including its surface expression level and the repertoire of peptides bound [48].

It is unclear whether *HLA-E* alleles are expressed co-dominantly or not, however, they probably are, by analogy with other related genes of MHC class I locus, like *HLA-A, -B, -C*, or *MIC*. Homozygous for *HLA-E^G^* cells have significantly higher affinity to peptides and higher surface expression level of HLA-E than cells homozygous for *HLA-E^R^* [48,49]. Thermal stability of the protein is also higher in the case of *HLA-E^G^* [48]. Herewith, both allelic proteins in complexes with peptides bind their CD94/NKG2x receptors with close affinity [40]. Together, presented differences suggest that in individuals, homozygous for *HLA-E^R^*, NK cell activation, or inhibition via NKG2x receptor will be lower than in those for *HLA-E^G^* [36].

The *HLA-E^G^* allele, but not *HLA-E^R^*, was found in genomes of some ape species, indicating that *HLA-E^G^* is more ancient. The emergence of both of these alleles seems to happen prior to the resettlement of humans from Africa and separation of races. Their maintenance was supported by balancing selection, evidencing for being advantageous for individuals which are heterozygous at the HLA-E gene locus [50]. It is also possible that substitution of *HLA-E^G^* allele by *HLA-E^R^* is taking place throughout evolution, and the current distribution of both alleles will change in favor of the latter [50]. Either way, as far as *HLA-E^R^* appeared later than *HLA-E^G^* and now its frequency has reached about 50%, it seems to be advantageous for the human population. To analyze the reason for this, we will review the *HLA-E* alleles association with the different diseases, including viral and bacterial infections, cancer, and autoimmune disorders, in the sections below.

## 5. Repertoire of Peptides Presented by HLA-E in Non-Typical Conditions

As mentioned above, in certain conditions the “normal” repertoire of peptides presented in complex with HLA-E (i.e., peptides derived from signal sequences of HLA-A, -B, -C, and -G) can be expanded by peptide fragments derived from stress proteins, viral, or bacterial proteins. In some cases, that precludes the NK cells NKG2A-based tolerance, e.g., Hsp60 derived peptide complexes with HLA-E interfere with binding with either NKG2A or NKG2C, that impairs the NKG2A-mediated inhibition of NK cell lysis [7]. In other cases, such alternative peptides serve for maintenance of NK cell inhibiting via engaging of NKG2A receptors. The examples are infections of HCMV [33], vaccinia virus [9], HIV [51], HCV [52]. All these virus-derived peptides inhibit NK cell-mediated lysis of infected cells. So, it can be postulated that during evolution viruses acquired these features, allowing them to evade NK cells surveillance.

Often in conditions of viral infections or in a number of malignancies, expression of classical MHC class I proteins is abrogated by inhibition of TAP transporter and there are no HLA-derived peptides to present in complex with HLA-E. In this case, the repertoire of peptides associated with HLA-E drastically changes. In the model of TAP-deficient human cells, 552 peptides of different length (8- to 13-mer) and sequences presented by HLA-E were detected [53]. Another study showed 36 peptides able to form complex with HLA-E in the absence of HLA class I expression. They had up to 16 amino acid residues and substantially differed from conventional HLA-derived peptides. Also, in contrast with these peptides, unusual peptides did not protect cells from NK cell-mediated lysis [54]. Interestingly, the peptide repertoires presented by HLA-E molecules of different alleles in conditions of absence of expression of classical HLA I do not overlap [55]. The biological significance of this finding is to be elucidated.

## 6. Soluble HLA-E and its Disease Associations

HLA-E is, in particular, expressed on endothelial cells. Here, under inflammatory conditions, cytokines such as tumor necrosis factor α (TNF-α), interleukin-1β (IL-1β), and interferon-γ (IFN-γ) up-regulate cell surface expression and, moreover, the release of soluble HLA-E (sHLA-E) molecules. It is considered that a high level of HLA-E protects endothelial cells from NK cell lysis, while sHLA-E protects bystander cells [15]. The release of sHLA-E is performed by cleavage with metalloproteinases [56]. Using Western blotting, sHLA-E was shown to be absent in normal sera, but presented in sera of patients in active phase of ANCA–associated systemic vasculitis [15]. Using enzyme-linked immunosorbent assay (ELISA), it was shown that sHLA-E was presented in healthy donors, but its level was relatively higher in plasma samples of patients with melanoma [57], neuroblastoma [58], and chronic lymphocytic leukemia [59] on advanced disease stage. In addition, Takayasu arteritis patients on active stages demonstrated significantly higher level of HLA-E than those with stable disease [60]. Thus, sHLA-E is presented in sera of healthy individuals, and increase of its level correlates with a number of disorders, primarily cancer and autoimmune diseases. However, more research is needed to further understand the role of the soluble form of HLA–E in aforementioned disorders.

## 7. HLA-E Dimorphism and Viral Infections

Patients with chronic hepatitis C infected with HCV genotype 2 or 3 were significantly less likely to display the *HLA-E^R^/HLA-E^R^* genotype than healthy control group [61]. The study performed on a cohort of HIV/HCV co-infected patients confirmed this observation [62]. This result can be explained by upregulation of HCV on NKG2A expression on NK cells; accordingly, presence of the HLA-E allele, which produces protein with higher surface expression *(*01:03)*, could lead to higher inhibition of NK cells and less effective immune response, resulting in a chronic infection [63]. Also, homozygosity for HLA-E^R^ has been shown to be a protection factor for BK polyomavirus (BKPyV) induced nephropathy following kidney transplantation [64].

On the other hand, one more group, which also investigated patients co-infected with HIV and HCV, has revealed that *HLA-E^R^* is directly correlated with HCV co-infection [65]. Also, *HLA-E^R^* was found to be associated with hepatitis B or hepatitis B complicated with hepatocellular carcinoma [66]. Individuals homozygous for *HLA-E^G^ (*01:03)* allele have 3.5-fold decrease of HIV-1 infection risk in comparison with those who are heterozygous or homozygous for *HLA-E^R^ (*01:01)* [67].

The situation with HCMV should be separately mentioned. Nearly 100% of the worldwide human population is infected with HCMV, and no association between HLA-E alleles and the presence of HCMV infection in the organism was found. Equal distribution of HLA-E alleles for HCMV^–^ and HCMV^+^ hosts was found [42]. However, there were differences concerning activity of the virus in immunocompromised patients, in particular, in conditions following solid organ transplantation. At the current moment, only one research in this area has been performed: Guberina et al. have shown that *HLA-E^G^* allele is associated with HCMV reactivation in patients following kidney transplantation [68]. Interestingly, no preference in homozygous state of any HLA-E allele associated with the establishment of an HLA-E*UL40 CD8^+^ T cell response has been found. Instead, Jouand and colleagues reported a higher prevalence of *HLA-E*01:01/01:03* heterozygotes among individuals manifesting CD8^+^ T cell response to HLA-E presenting UL40-derived peptide [42].

No association of HLA-E alleles with HPV infection has been found [69]. Thus, the HLA-E alleles’ association with different viral infections seems to be discrepant. Each of the alleles can be advantageous for an individual or the entire population against a particular virus. Therefore, we can assume that the presence of both alleles in the gene pool can be beneficial for the survival of the population.

## 8. HLA-E Dimorphism and Bacterial Infections

There are several studies concerning associations of HLA-E alleles with bacterial infections. Prasetyo and co-workers found, *HLA-E^R^* association with Toxoplasma gondii co-infection in a cohort of HIV-infected patients [65]. Tamouza and coallegues (2007) found that the presence of *HLA-E^G^* allele in genotype decreased the risk of severe bacterial infections in sickle cell anemia individuals [70]. The same group found *HLA-E^R^* to be a risk factor for severe bacterial infections and transplant-related mortality in a cohort of patients who had undergone bone marrow transplantation [71]. Taken together, these works suggest the protecting role of *HLA-E^G^* allele in bacterial infections.

## 9. HLA-E Dimorphism and Cancer

It is shown that presence of *HLA-E^G^ (*01:03)* allele is associated with higher level of sHLA-E and increased risk of mortality in chronic lymphocytic leukemia (CLL) [59]. Women carrying the *HLA-E^G^* allele are more susceptible to serous ovarian cancer [72]. *HLA-E^G^* homozygous genotype is associated with significantly higher risk of acute leukemia and higher level of sHLA-E in the patient group [73]. Among nasopharyngeal carcinoma patients, *HLA-E^G^* allele was significantly more frequent than *HLA-E^R^* [74]. These results have been supported by another work [75]. At last, genotypes with at least one copy of *HLA-E^R^* are associated with lower risk of EBV-related classical Hodgkin’s lymphoma [76].

However, in a number of research studies, no association of HLA-E alleles with cancer has been reported. One more study of nasopharyngeal carcinoma patients revealed no significant difference between both alleles [77]. No associations were found between HLA-E alleles and urinary bladder carcinoma [78].

Primary cutaneous melanoma cases showed no association with any of the HLA-E alleles [79]. HLA-E alleles did not affect the clinical outcome of patients with stage III colorectal cancer [80].

It is important to notice that HLA-E is upregulated in melanoma [81], colorectal cancer [82,83,84,85], non-small cell lung carcinoma [86], serous ovarian carcinoma [87], and gynecological cancers [88], suggesting that HLA-E plays an important role in the progression of these malignancies, probably, by inhibiting NKG2A^+^ killer lymphocytes. We can suggest that high surface level of HLA-E on cancer cells makes the differences between both alleles negligible in these diseases.

Taken together, these data indicate that, although some malignancies do not seem to correlate with a presence of any of the HLA-E alleles, the substantial number of cancer diseases is associated with *HLA-E^G^ (*01:03)* allele. No association between cancer and *HLA-E^R^* allele has been reported. Hirankarn with colleagues suggested that *HLA-E^R^* could be beneficial for antitumor immune responses due to lower surface expression of *HLA-E*01:01,* which results in lower inhibitory action for CD94^+^/NKG2A^+^ immune cells [74]. Also, two reports indicated the association between *HLA-E^G^* and the level of sHLA-E in patients’ sera [59,73]. As discussed earlier, sHLA-E provides the protection of cells from NK-mediated lysis. Therefore, this can be potential mechanism for explaining the role of *HLA-E^G^* allele as a risk factor for a number of cancer diseases. However, to prove it the wide screening of different cancer diseases for HLA-E genotype and sHLA-E level is needed.

## 10. HLA-E Dimorphism and Autoimmune Diseases

*HLA-E^R^* was found to be associated with type 1 diabetes mellitus [89] and ankylosing spondylitis [90]. Later, another group reported that none of the alleles were associated with ankylosing spondylitis, though heterozygous phenotype showed protective effect [91]. The controversial data were provided by groups investigating pemphigus vulgaris: According to one publication, *HLA-E^G^* homozygotes are associated with this disease [92], according to another one, *HLA-E^G^*/*HLA-E^R^* heterozygous genotype increased the risk of the disorder [93]. In addition, *HLA-E^R^*/*HLA-E^R^* genotype is associated with reduced risk of rheumatoid arthritis [94]. *HLA-E^G^* homozygous individuals tend to have lower risk of psoriatic arthritis [95]. This observation is in agreement with the study performed on patients with psoriasis, where they showed that homozygotes for *HLA-E^R^* allele had a higher risk of this disease [96].

Thus, it seems that according to currently available data, there is no clear association of any HLA-E allele with autoimmune disorders. Therefore, one would expect that *HLA-E^G^* genotype lowers the risk of such diseases due to higher surface expression of its allele product and, consequently, higher inhibitory effect on NKG2A/B^+^ cytotoxic lymphocytes. However, apparently there is no tight association between HLA-E surface expression and inhibitory effect of such cytotoxic cell subpopulations. It is important to note that the ligand-receptor system of HLA-E and NKG2A/B seems to play the considerable role in a number of autoimmune diseases, including rheumatoid arthritis [97,98,99], ankylosing spondylitis [100], and psoriasis [101,102,103].

## 11. HLA-E Dimorphism and Transplantation

We will also consider the association of *HLA-E* alleles with clinical outcome after both solid organs and hematopoietic stem cells (HSC) or bone marrow transplantation. At the moment there are only two studies published concerning solid organ transplantation. Di Cristofaro and co-workers have found that *HLA-E^G^* correlates with higher chronic lung allograft dysfunction in comparison with patients homozygous for *HLA-E^R^* allele [104]. Guberina and colleagues found the association of *HLA-E^G^* allele with HCMV recurrence in patients following kidney transplantation [68]. Thus, little information is known, and more research is needed to further study the role of *HLA-E* and its association with clinical outcome of patients following solid organ transplantation.

However, more is known about influence of *HLA-E* alleles on HSC and bone marrow transplantation outcome. It was shown that acute graft versus host disease and risk for transplant-related mortality was lower, and overall survival was higher in cases of HSC transplantation, when donor and recipient had matched homozygous *HLA-E^G^* genotype [105]. The same group indirectly confirmed this result by finding that homozygous state of donors for *HLA-E^R^* was a risk factor for early severe bacterial infections and transplant-related mortality in bone marrow transplantation [71]. Also, transplantation of HSC from donors, homozygous for *HLA-E^G^*, to HLA-matched recipients was shown to correlate with higher overall survival and lower transplant-related mortality [106]. Together, these three works have shown the positive role of *HLA-E^G^* allele in bone marrow and HSC transplantation. However, at the moment we cannot count these results enough reliable: One more work has shown only a slight trend to this direction [107], and the last one did not find any correlation between current HLA-E allele and clinical outcome of HSC transplantation [108].

Thus, to date, it seems that *HLA-E^G^* allele is beneficial for HSC transplantation, however, more studies is required to confirm this statement.

## 12. HLA-E and Pregnancy Disorders

The presence of *HLA-E*01:01* allele in genotype was found to be associated with recurrent abortions in Egyptian women [109] and in a group of Indian women [110]. However, another work indicated that *HLA-E* alleles do not influence frequencies of recurrent abortions in a cohort of Japanese women [111]. Also, no association with *HLA-E* alleles was observed in a group of Danish women [112]. Meta-analysis of data from studies indicated above revealed the increased risk of recurrent miscarriage in mothers carrying an *HLA-E*01:01* allele [113]. Taken together, *HLA-E*01:01* allele is associated with higher frequency of recurrent abortions. The probable reason for this, in our opinion, can be lower inhibitory action of HLA-E protein encoded by this allele on NK cells in comparison with allele *HLA-E*01:03.*

Also, there is a single report concerning the role of HLA-E in successful natural conception and pregnancy. The *HLA-E*01:03* allele was found to be significantly increased in men who needed assisted reproduction treatment as compared to fertile men, suggesting that this allele may be disadvantageous in natural conception and pregnancy outcome, whereas the *E*01:01* allele may manifest a protective effect [114].

## 13. Concluding Remarks

The data about association of *HLA-E* alleles with different pathological states reviewed above are summarized in the Table 2. Despite the lower affinity to peptides, lower surface expression, and thermal stability of the corresponding protein, allele *HLA-E^R^* manifests the clear advantages for individuals subjected to a number of diseases. Probably this can be linked to a repertoire of peptides associated with HLA-E protein and its lower inhibitory effect on cytotoxic lymphocytes as, for example, in the case of cancer diseases.

Although worldwide the *HLA-E* alleles are presented in nearly equal frequencies, the specific human populations demonstrate quite different ratios of HLA-E alleles frequencies. For example, in Japan, the frequency of *HLA-E^R^* is about 32% (*HLA-E^G^*, accordingly, 68%), whereas in India (Lucknow)—70.4% of *HLA-E^R^* and 29.6% of *HLA-E^G^* (http://www.allelefrequencies.net/hla6006a.asp?hla_locus_type = Non-Classical, locus E, populations Japan pop13 and India Lucknow). Such discrepancies may be linked not only to ethnic genetic specificities, but also to different epidemiologic situations in various countries. These factors can shift the ratios of *HLA-E* alleles in certain population to one or another direction. Taken together, both alleles can be beneficial for the whole human population, and this can be the reason for these alleles to have been supported by balancing selection.

## Figures and Tables

**Figure 1 ijms-20-05496-f001:**
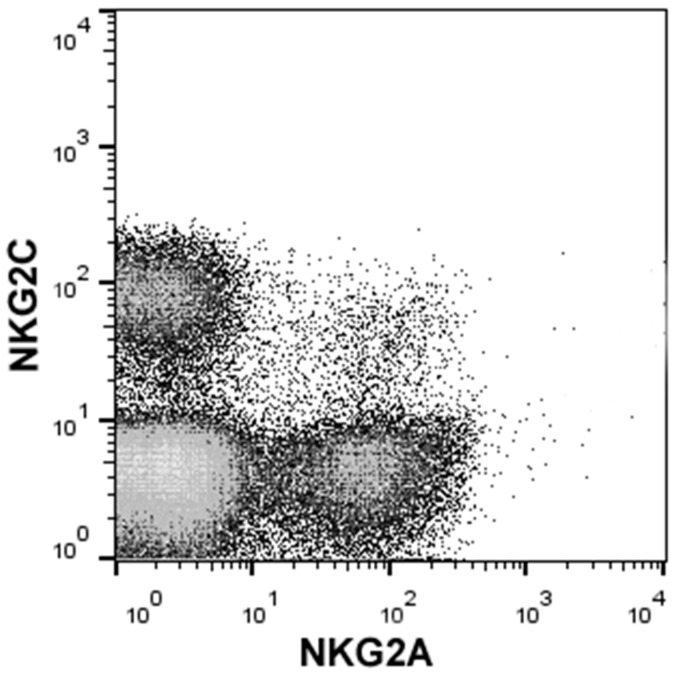
NK cells, freshly isolated from human peripheral blood by magnetic separation, stained with fluorescent-labeled antibodies to NKG2A and NKG2C with gating on CD3^–^CD56^+^ lymphocytes.

**Table 1 ijms-20-05496-t001:** Peptides, derived from leader sequences of HLA class I proteins, endogenous proteins Hsp60 and MRP7, and pathogen-derived peptides, presented by HLA-E. From Lauterbach et al. [11], and Sharpe et al. [12], with modifications.

VMAPRTLLL	HLA-A*01, -A*03, -A*11, -A*29, -A*30, -A*31, -A*32, -A*33, -A*36, and -A*74, HLA-C*02 and C*15
VMAPRTLVL	HLA-A*02, -A*23, -A*24, -A*25, -A*26, -A*34:02, -A*34:06, -A*43, -A*66, and -A*69
IMAPRTLVL	HLA-A*34:01
VMPPRTLLL	HLA-A*80
VMAPRTVLL	HLA-B*07, -B*08, -B*14, -B*38, -B*39, -B*42, -B48, -B*67, -B*73 and -B*81
VTAPRTLLL	HLA-B*13, -B*18, -B*27, -B*37, -B*40, -B*44, -B*47, -B*54, -B*55 -B*56, -B*59, -B*82 and -B*83
VTAPRTVLL	HLA-B*15, -B*35, -B*40, -B*41, -B*44:18, -B*45, -B*46, -B*49, -B*50, -B*51, -B*52, -B*53, -B*57, -B*58 and -B*78
VMAPRTLIL	HLA-C*01, -C*03, -C*04, -C*05, -C*06, -C*08, -C*12, -C*14, -C*16 and -C*17:02
VMAPRALLL	HLA-C*06:17, -C*07 and -C*18
VMAPRTLTL	HLA-C*08:09
VMAPQALLL	HLA-C*17:01, C*17:03 and -C*17:05
VMAPRTLFL	HLA-G*01
QMRPVSRVL	Hsp60
ALALVRMLI	ATP-binding cassette protein MRP7
VMAPRTL(I/V/L)L	HCMV UL40 protein
YLLPRRGPRL	Hepatitis C virus core protein
SQAPLPCVL	Epstein–Barr virus BZLF1 protein
AISPRTLNA	HIV, P24
RMAATAQVL	*Mycobacterium tuberculosis,* Mtb14
RMPPLGHEL	*Mycobacterium tuberculosis,* P49
RLPAKAPLL	*Mycobacterium tuberculosis,* Mtb44
GMQFDRGYL	*Salmonella typhimurium* serovar Typhi, GroEL

**Table 2 ijms-20-05496-t002:** The role of *HLA-E* alleles, **01:01* and **01:03*, in different diseases.

Pathologies	*HLA-E^R^ (*01:01)*	*HLA-E^G^ (*01:03)*
Viral infections	protection/risk	protection/risk
Cancer ^1^	protection	risk
Bacterial infections	risk	protection
HSC and bone marrow transplantation ^2^	risk	protection
Recurrent abortions	risk	protection
Autoimmune disorders	protection/risk	protection/risk

^1^ In a part of malignant diseases no association with any of the alleles has been found; ^2^ There is only trend to this regularity; more investigation is needed to confirm it.

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
