# Peer review of "Dimorphism of HLA-E and Its Disease Association"

_ijms, 2019, doi:10.3390/ijms20215496_

Round 1

Reviewer 1 Report

I enjoyed reading this review of Kanevskiy et al. on the dimorphism of HLA-E and links with disease. I have some suggestions to improve the manuscript:

Throughout the manuscript, the authors have not included a discussion of HLA-E restricted T cells in disease outcome. Although the authors chose to focus on NKG2x/CD94 receptors, the potential contribution of these T cells may be important in disease outcome and should be discussed. There is no reference for Figure 1 and/or the legend is lacking information. If Figure 1 is data generated by the authors, please indicate what was done and what is shown in Figure 1 (I assume gated on CD56+ CD3- lymphocytes but should be stated). The section of Evolution and polymorphism is out of place. This should be earlier in the manuscript. HLA-E dimorphism and viral infection is lacking references on susceptibility to CMV e.g. Guberina et al. J Infect Dis. 2018, 217:1918-1922 The allelic frequencies (page 7 line 290-294) is out of place and should appear in the section on polymorphism Page 1 line 36-41 should be referenced The manuscript has poor English in parts e.g. Page 6 line 203 “Anyway” should be avoided Page 6 line 211 “stimulating action” should be “upregulation” A typo on line 178 “thepressure” should be 2 words

Author Response

Point 0: I enjoyed reading this review of Kanevskiy et al. on the dimorphism of HLA-E and links with disease. 

Response 0: Thank you very much for your interest!

Point 1: Throughout the manuscript, the authors have not included a discussion of HLA-E restricted T cells in disease outcome. Although the authors chose to focus on NKG2x/CD94 receptors, the potential contribution of these T cells may be important in disease outcome and should be discussed.

Response 1: The section about HLA-E restricted T cells has been included. Page 4, line 145

Point 2: There is no reference for Figure 1 and/or the legend is lacking information. If Figure 1 is data generated by the authors, please indicate what was done and what is shown in Figure 1 (I assume gated on CD56+ CD3- lymphocytes but should be stated).

Response 2: Yes, this is our data. We have supplemented the legend with appropriate information. Page 4, lines 122-124

Point 3: The section of Evolution and polymorphism is out of place. This should be earlier in the manuscript.

Response 3: The section has been replaced. Page 5, line 168

Point 4: HLA-E dimorphism and viral infection is lacking references on susceptibility to CMV e.g. Guberina et al. J Infect Dis. 2018, 217:1918-1922  

Response 4: The missing fragment has been included. Page 7, line 253

Point 5: The allelic frequencies (page 7 line 290-294) is out of place and should appear in the section on polymorphism

Response 5: We have included allelic frequencies and discussion about proportions of alleles in different countries in section of polymorphism. Page 5, lines 174-177

Point 6: Page 1 line 36-41 should be referenced

Response 6: The reference has been added.

Point 7: The manuscript has poor English in parts e.g. Page 6 line 203 “Anyway” should be avoided Page 6 line 211 “stimulating action” should be “upregulation” A typo on line 178 “thepressure” should be 2 words 

Response 7: These mistakes have been corrected. “Anyway” has been changed at the page 5 line 198, “upregulation” – page 6 line 243, “the pressure” – page 5, line 172. Also, English was corrected by English speaking colleague

Reviewer 2 Report

It is an interesting summary of the current knowledge about the HLA-E dimorphisms and disease association. However some aspect are missing: (i) HLA-E dimorphism and viral infections: Allelic association with CMV infection is missing, (ii) HLA-E dimorphism in solid transplantation- and alloSCT-related complication are missing; (iii) HLA-E dimorphism and pregnancy disorders are missing. These points should be included. In some cases (see below) only review article are given and not the original publication are mentioned. Sometimes references are missing, or reports of are imprecise mentioned. Please improve these issues.

It differs from classical (MHC Ia) proteins by limited polymorphism and specific set of presented peptides [1]. Review article please include the original article

A few HLA-C allotypes and the leader peptides from HLA-F and HLA-E itself that do not match this motif [3]. Review article include the original article(s)

Expression of HLA-E is often detected on cells of different tissues in pathologic conditions, such as cancer, bone marrow transplantation, autoimmune diseases [8,15,16].no ref. for autoimmune disease are given

In the absence of MHC Ia expression there are no peptides to present with HLA-E, and HLA-E becomes downregulated. In this case NK cells do not receive the inhibitory signal and can lyse the target cell using a number of activating receptors, which recognize a set of ligands expressed in conditions of infection, cell stress or malignization of the cell [21] Review article please include a original article that report that in case of HLA-I a lack HLA-E is downregulated as not HLA-Ia leader peptide is provided

sHLA-E is absent in normal sera, but it presence in plasma correlates with a number of disorders, primarily cancer and autoimmune diseases, including ANCA–associated systemic vasculitis [13], Takayasu arteritis [43], melanoma [44], chronic lymphocytic leukemia [45] and neuroblastoma [46]. The information about soluble form of HLA-E is still very poor. This is not correct: by immunoprecipitation studies sHLA-E is not detectable in blood samples of healthy controls (13), but by quantifying sHLA-E via ELISA substantial amounts of sHLA-E levels are detectable (44,45, 46) in healthy controls. Please edit this issue.

Author Response

Point 1: HLA-E dimorphism and viral infections: Allelic association with CMV infection is missing  

Response 1: The missing fragment has been included. Page 7, line 253

Point 2: HLA-E dimorphism in solid transplantation- and alloSCT-related complication are missing

Response 2: The missing section has been added. Page 8, line 328

Point 3: HLA-E dimorphism and pregnancy disorders are missing.

Response 3: The appropriate section has been added. Page 10, line 353

Point 4: It differs from classical (MHC Ia) proteins by limited polymorphism and specific set of presented peptides [1]. Review article please include the original article

Response 4: The reference to the original article has been inserted. Page 1, line 31

Point 5: A few HLA-C allotypes and the leader peptides from HLA-F and HLA-E itself that do not match this motif [3]. Review article include the original article(s)

Response 5: The references to the original articles have been inserted. Page 1, line 36

Point 6: Expression of HLA-E is often detected on cells of different tissues in pathologic conditions, such as cancer, bone marrow transplantation, autoimmune diseases [8,15,16].no ref. for autoimmune disease are given

Response 6: The required references have been added. Page 3, line 59.

Point 7: In the absence of MHC Ia expression there are no peptides to present with HLA-E, and HLA-E becomes downregulated. In this case NK cells do not receive the inhibitory signal and can lyse the target cell using a number of activating receptors, which recognize a set of ligands expressed in conditions of infection, cell stress or malignization of the cell [21] Review article please include a original article that report that in case of HLA-I a lack HLA-E is downregulated as not HLA-Ia leader peptide is provided

Response 7: The proper reference has been added. Page 3, line 82

Point 8: sHLA-E is absent in normal sera, but it presence in plasma correlates with a number of disorders, primarily cancer and autoimmune diseases, including ANCA–associated systemic vasculitis [13], Takayasu arteritis [43], melanoma [44], chronic lymphocytic leukemia [45] and neuroblastoma [46]. The information about soluble form of HLA-E is still very poor. This is not correct: by immunoprecipitation studies sHLA-E is not detectable in blood samples of healthy controls (13), but by quantifying sHLA-E via ELISA substantial amounts of sHLA-E levels are detectable (44,45, 46) in healthy controls. Please edit this issue.

Response 8: This fragment has been rephrased. Page 7, lines 230-238

Round 2

Reviewer 2 Report

No further comments.